# A New Method for Improving Extraction Efficiency and Purity of Urine and Plasma Cell-Free DNA

**DOI:** 10.3390/diagnostics11040650

**Published:** 2021-04-03

**Authors:** Selena Y. Lin, Yue Luo, Matthew M. Marshall, Barbara J. Johnson, Sung R. Park, Zhili Wang, Ying-Hsiu Su

**Affiliations:** 1JBS Science, Inc., Doylestown, PA 18902, USA; slin@jbs-science.com (S.Y.L.); mmarshall@jbs-science.com (M.M.M.); bjohnson@jbs-science.com (B.J.J.); zwang@jbs-science.com (Z.W.); 2The Baruch S Blumberg Institute, Doylestown, PA 18902, USA; yueluo98@yahoo.com (Y.L.); sungryeolpark0906@gmail.com (S.R.P.)

**Keywords:** cell-free DNA, biospecimen variability, liquid biopsy, plasma, urine, precision medicine

## Abstract

This study assessed three commercially available cell-free DNA (cfDNA) extraction kits and the impact of a PEG-based DNA cleanup procedure (DNApure) on cfDNA quality and yield. Six normal donor urine and plasma samples and specimens from four pregnant (PG) women carrying male fetuses underwent extractions with the JBS cfDNA extraction kit (kit J), MagMAX Cell-Free DNA Extraction kit (kit M), and QIAamp Circulating Nucleic Acid Kit (kit Q). Recovery of a PCR product spike-in, endogenous TP53, and Y-chromosome DNA was used to assess kit performance. Nucleosomal-sized DNA profiles varied among the kits, with prominent multi-nucleosomal-sized peaks present in urine and plasma DNA isolated by kits J and M only. Kit J recovered significantly more spike-in DNA than did kits M or Q (*p* < 0.001) from urine, and similar amounts from plasma (*p* = 0.12). Applying DNApure to kit M- and Q-isolated DNA significantly improved the amplification efficiency of spike-in DNA from urine (*p* < 0.001) and plasma (*p* ≤ 0.013). Furthermore, kit J isolated significantly more Y-chromosome DNA from PG urine compared to kit Q (*p* = 0.05). We demonstrate that DNApure can provide an efficient means of improving the yield and purity of cfDNA and minimize the effects of pre-analytical biospecimen variability on liquid biopsy assay performance.

## 1. Introduction

Liquid biopsies present a minimally invasive or noninvasive approach for detecting circulating cell-free DNA (cfDNA) markers in prenatal genetic testing [1,2,3,4]. The cfDNA isolated from plasma and urine is often low in quantity and highly fragmented as it is mostly derived from apoptotic cells. Fragments of cfDNA range predominantly from 130 to 250 bp, reflecting the mono-nucleosomal-sized DNA that is the major cfDNA species in plasma [5,6,7] and urine [8], and are even smaller in the case of circulating cell-free fetal DNA (ccffDNA) in maternal urine [4]. While cfDNA is a promising analyte for liquid biopsy, detection of cfDNA markers is significantly impacted not only by pre-analytical variables such as cfDNA isolation [9], but also by biospecimen variability.

Platforms for cfDNA isolation, which are primarily column- or magnetic-bead-based, can be impacted by this variability, including the levels of biologically active molecules, overall molecular composition, and the presence of impurities. This phenomenon in turn results in variable DNA yield, quality, and residual impurity content. Urine is known to be a highly complex sample matrix with high inter- and intra-person variability [10]. In an effort to reduce the impact of biospecimen variability on cfDNA isolation and subsequent analytical performance, we developed a PEG-based DNA cleanup step (DNApure) to remove impurities from extracted cfDNA and cell-associated large genomic DNA.

In this study, we compared the analytical performance of two frequently used, commercially available kits, the bead-based MagMAX Cell-free DNA extraction kit and the column-based QIAamp Circulating Nucleic Acid kit, with that of the JBS Science cfDNA extraction kit, developed by our laboratory, before and after DNApure cleanup. We compared cfDNA fragment size, yield, purity (as assessed by PCR amplification efficiency), and their reproducibility among the three kits in normal donor urine and plasma and in specimens collected from pregnant (PG) donors carrying a male fetus. Our results demonstrate that the PEG-based DNApure cleanup procedure improves the purity of cfDNA isolated from urine and plasma by all three kits and highlight the importance of obtaining large quantities of high-quality cfDNA in improving downstream liquid biopsy applications.

## 2. Materials and Methods

### 2.1. Urine and Plasma Sample Collection

All human specimens used in this study were purchased from BioIVT^®^ (Westbury, NY, USA) or Lee Biosolution (Maryland Heights, MO, USA) or were obtained as archived and de-identified samples from other studies as described previously [8]. The only patient characteristic obtained was sex (Appendix A). Frozen human urine (50 mL) was collected from healthy female (*n* = 3) and male (*n* = 3) volunteers and mixed with EDTA to achieve 30–50 mM final EDTA concentration. Normal donor blood collected in K_2_EDTA tubes was obtained from BioIVT^®^. Plasma was separated from whole blood by centrifugation at 2800× g for 20 min and filtered (0.2 µm pore size) in 2.0 mL aliquots. Plasma and urine specimens were collected from four PG females in their third trimester carrying a male fetus. The first PG donor had matched urine and plasma collected at two different times, T1 and T2, and three subsequent plasma-only collections (PT3, PT4, and PT5). PG donor 2 provided only urine samples collected at three different times, T1, T2, and T3. PG donors 3 and 4 had only plasma obtained by Lee Biosolution. In this case, blood was collected by venipuncture and centrifuged at 3500 RPM; the plasma was then pipetted off and stored at −20 °C. The specimens used in this study are shown in Figure 1. All experiments were performed in accordance with relevant guidelines and regulations.

### 2.2. cfDNA Extraction

The cfDNA was extracted from normal urine and plasma samples in 3–4 replicates according to the manufacturers’ protocols for the following three kits: the JBS DNA extraction kit, which includes PEG-based DNApure cleanup (kit J; cat# JBS-08872 for urine or cat# JBS-08874 for plasma; JBS Science, Doylestown, PA, USA), the MagMAX Cell-Free DNA Extraction (kit M; cat# A29319, Thermo Fisher Scientific, Waltham, MA, USA), and the QIAamp Circulating Nucleic Acid kit (kit Q; cat# 55114, Qiagen, Germantown, MD, USA). Of note, both kit M and kit Q are applicable for both urine and plasma.

Each urine/plasma specimen was pooled, mixed well, and aliquoted for extraction. Extraction was performed from 3.0 mL urine or 2.0 mL plasma aliquots in triplicate with each kit. Kit M and Q urine and plasma samples were ultra-centrifuged at 16,000× *g* for 10 minutes at 4 °C prior to cfDNA extraction. Aliquots were immediately extracted or frozen at −20 °C until cfDNA extraction. Due to the low amounts of fetal DNA found in maternal urine, PG urine samples were extracted using up to 4.0 mL urine inputs. Due to the limited quantities of PG plasma obtained from Lee Biosolution, extraction of 0.5 mL plasma aliquots was performed for kit J and Q assessment only.

### 2.3. Synthetic 141 bp Spike-In DNA

Normal urine and plasma biopsies were spiked with a PCR-produced 141 bp double-stranded DNA fragment at 10^6^ copies/mL prior to cfDNA extraction. The synthetic spike-in DNA was quantified using the JBS Artificial Spike-In DNA Quantification kit according to the manufacturer’s specifications, and the recovered copies were estimated in each cfDNA sample. We calculated spike-in recovery as
Total Output/Total Input=% recovery

### 2.4. Size Assessment

We visualized cfDNA profiles on High Sensitivity D5000 ScreenTapes and a TapeStation 4200 system (Agilent Technologies, Santa Clara, CA, USA). The equivalent of 0.08 mL urine or plasma was visualized for each specimen.

### 2.5. Quantification Assays

*TP53* gene quantity was measured by qPCR as previously described [11], and Y-chromosome DNA was measured using the Y-Chromosome DNA quantification kit (JBS Science). To assess the amounts of residual protein impurities after cfDNA isolation, the protein concentration of each sample was determined using the Qubit Protein Assay kit (cat# Q33211, Thermo Fisher Scientific) with input equivalents of 0.08 mL urine and 0.08–0.20 mL plasma. Quantitative PCR assays were performed in duplicate on the LightCycler 480 platform (Roche, Indianapolis, IN, USA).

### 2.6. Statistical Analyses

To compare the three DNA isolation kits, each of the liquid biopsies from 6 normal urine and plasma donors were spiked with synthetic 141 bp DNA, mixed, and split into three replicates. DNA from each of the aliquots was isolated as three independent specimens with each kit to obtain 18 data sets for comparison. We tested data for normality using the Shapiro–Wilk test and for homogeneity of variance using Levene’s test. We tested the hypothesis of no difference among DNA extraction kits J, M, and Q in recovery of synthetic 141 bp or *TP53* DNA using a one-way analysis of variance (ANOVA), or a Kruskal–Wallis test when parametric assumptions were not met. For post hoc comparisons among DNA extraction kits, we used Tukey’s post hoc comparison for ANOVA procedures. We tested the hypothesis of no difference in recovery of synthetic 141 bp DNA in samples extracted with kits M and Q before and after conducting the DNApure cleanup procedure using a paired *t*-test or Wilcoxon rank sum test when parametric assumptions were not met. For all statistical tests, we used α = 0.05 to determine whether differences were significant. All analyses were conducted in R v3.6.1 (R Development Core Team, 2013).

## 3. Results

### 3.1. Comparison of cfDNA Extraction Kit Performance in Urine

Given the importance of obtaining nucleosomal-sized DNA from liquid biopsies, we first compared the overall fragment size distributions of cfDNA isolated from six normal urine donors with the three kits. Because it is known that sex impacts urinary DNA yield [12], three male and three female donors were included (Figure 1A). Representative electropherograms of cfDNA isolates obtained with each kit are compiled for each donor in Figure 2. Interestingly, in some donors the DNA size profiles were different among the kits. For example, donors 1 and 6 showed detectable mono- and di-nucleosomal-sized peaks in kit J and M, but not in kit Q isolates. Overall, all three kits were able to isolate mono-nucleosomal-sized DNA, except in male donor 5, whose samples produced low DNA yields with all three kits.

Impurities, salt, or the amount of large genomic DNA in the isolated cfDNA could impact the size distribution and peak heights in the electropherogram analysis. As mono-nucleosomal-sized DNA is the predominant species of cfDNA, we evaluated the efficiency of the three kits in recovering mono-nucleosomal-sized DNA. We utilized a 141 bp double-stranded PCR product spike-in as a mimic of mono-nucleosomal-sized cfDNA to assess recovery efficiency. The PCR product was added into each urine sample at 10^6^ copies/mL. As assessed by qPCR, the spike-in DNA was recovered at concentrations ranging from 7.8 × 10^4^ to 3.7 × 10^6^ copies/mL among the six donors. However, on average, kit J recovered significantly more spike-in DNA than did kits M or Q (ANOVA, *p* < 0.001) (Appendix A, Figure 3A). The apparent spike-in recovery efficiency, exceeding 100% in some samples, may indicate further spike-in PCR product purification during the extraction process, improving the downstream PCR amplification efficiency.

To better understand if the wide range of spike-in DNA recovery was due to impurities/PCR inhibitors in the eluted urine cfDNA, we measured the protein levels in all extracted urine cfDNA samples and found no detectable protein. Next, we applied the DNApure cleanup to urine DNA isolated with kits M and Q to assess whether it could improve PCR quantitation of the spike-in DNA. As shown in Figure 3B, a significant improvement of the PCR amplification efficiency of the spike-in DNA was achieved for both kits (paired *t*-test, *p* < 0.001) after the cleanup. The spike-in DNA recovery rates achieved with the DNApure procedure were comparable among all three kits (ANOVA, *p* = 0.343). Next, to assess the yields of urine cfDNA isolated with each kit after DNApure cleanup, we performed qPCR-based TP53 gene quantification in the extracted DNA. As expected, the urine TP53 levels ranged widely among the donors, from 1.77 × 10^2^ to 1.67 × 10^5^ copies/mL (Appendix A) and did not differ significantly among the kits (Kruskal–Wallis test, *p* = 0.23).

### 3.2. Performance of the JBS cfDNA Extraction Kit in Plasma

To assess if the PEG-based DNApure cleanup can also be applied to plasma cfDNA isolation, we next compared the performance of the three kits in normal plasma donors (Figure 1B), utilizing 2 mL of plasma for all extractions as described in Materials and Methods. Plasma cfDNA isolated from all six donors displayed a prominent mono-nucleosomal-sized peak when isolated with all three kits (Figure 4). Four donors displayed prominent di- and tri-nucleosomal DNA peaks in plasma cfDNA samples isolated by kits J and M; however, these peaks were undetectable or reduced in cfDNA isolated from the same donors by kit Q. The recovery of spike-in DNA did not differ significantly (Kruskal–Wallis test, *p* = 0.12) among the kits (Figure 5A). The PCR amplification efficiency of the recovered spike-in DNA significantly improved after the DNApure cleanup in samples extracted with both kit M (paired *t*-test, *p* = 0.013) and kit Q (paired *t*-test, *p* < 0.001) (Figure 5B). Interestingly, protein contamination was only detected in plasma cfDNA samples isolated with kit Q before cleanup (Appendix A) and was no longer detectable after cleanup (data not shown). Lastly, the *TP53* yield did not differ significantly (Kruskal–Wallis test, *p* = 0.61) among the kits (Appendix A) after DNApure cleanup. Overall, the plasma cfDNA extraction yield and quality were highly comparable among all three kits after cleanup, most likely because the PEG-based DNApure cleanup enabled the removal of PCR inhibitors and proteins co-purified with plasma cfDNA.

### 3.3. Extraction of ccffDNA in Liquid Biopsies

Some ccffDNA is released into the maternal circulation and can be detected in urine. Utilizing samples from four PG donors carrying a male fetus (Figure 1C), we assessed the efficiency of ccffDNA extraction from urine and plasma samples with the three kits according to the manufacturers’ protocols. A total of five urine specimens were collected from two third-trimester PG donors at various timepoints. We found the male Y-chromosome quantities to be significantly higher in kit J isolates as compared to kit Q isolates (Wilcoxon rank sum test, *p* = 0.05) and no significant differences were found between kits J and M (Wilcoxon rank sum test, *p* = 1.0) (Figure 6A). Due to the limited volumes of blood collected, only kits J and Q were evaluated on all seven plasma collections. As shown in Figure 6B, Y-chromosome quantities extracted from PG plasma were similar between the two kits.

## 4. Discussion

In this study, we compared the performance of the urine and plasma cfDNA isolation procedures embodied in three different kits and assessed the impact of the PEG-based DNApure method on cfDNA and ccffDNA (i.e., Y-chromosome DNA) yields in various normal and PG donor specimens. We demonstrated that DNApure cleanup can improve the quality of cfDNA isolated by both bead- and column-based commercial kits. All three isolation kits successfully extracted mono- and, frequently, di-nucleosomal-sized DNA. The JBS kit recovered significantly higher amounts of spike-in DNA and ccffDNA, as measured by qPCR assays, compared with the other kits. The PEG-based DNApure cleanup procedure can effectively remove impurities present in isolates obtained with the other kits, thus improving their downstream PCR amplification. This finding is highly significant since PCR amplification is a key methodology used in almost all liquid biopsy assay platforms.

The comparison of the three kits across six normal urine donors demonstrates the need for an extraction method capable of robustly handling a wide range of biospecimen variability. Biospecimen variability can stem from factors such as sex, disease type, and collection method and time. In this study, three male and three female urine donors were included as female urine is known generally to contain higher amounts of cfDNA than does male urine due to increased release of cfDNA from the female urinary tract. Consistent with this notion, samples from male donor 5 produced the lowest yields of urine cfDNA among all three kits. Interestingly, PG donor 1’s T1 and T2 urine samples were collected less than one week apart, but contained vastly different amounts of Y-chromosome ccffDNA, suggesting that multi-day specimen collection may be needed to increase detection sensitivity of cfDNA of interest. In addition, the three isolation kits utilize different technologies: kit Q uses a silica-column-based approach, which may capture larger fragments [12] compared with those if the magnetic-bead-based kits M and J. In the case of urine DNA isolation, the JBS method does not include a pre-processing centrifugation to avoid loss of cfDNA that may pellet with cellular debris (data not shown). Biospecimen variability among donors is further highlighted in the cfDNA size profiles of plasma samples extracted with the three kits. The visualization of nucleosomal-sized DNA by TapeStation is known to be impacted by salt/contaminants as well as the presence of large genomic DNA fragments. It is therefore possible the reduced di-and tri-nucleosomal-sized DNA peaks we observed in half of the plasma cfDNA samples extracted by kit Q may be caused by impurities. It is not surprising that PEG-based cleanup was able to remove proteins and PCR inhibitors in cfDNA extracted from liquid biopsy samples (Figure 3, Figure 5 and Appendix A), as PEG-based solutions have been used in previously developed DNA extraction and purification procedures [13,14]. The similar total amounts of cfDNA after cleanup, as assessed by *TP53* qPCR, obtained with all three kits are consistent with this explanation. Taken together, these observations highlight the value of the PEG-based DNApure cleanup in minimizing the impact of biospecimen variability on DNA isolation and, consequently, procuring high-quality cfDNA from liquid biopsy specimens.

As cfDNA in urine is less concentrated than in plasma [15], and large volumes of urine are often readily available, we developed our urine DNA isolation method to accommodate large input volumes (up to 50 mL), whereas the majority of commercial urine cfDNA extraction kits cannot process volumes more than 4 mL at a time. Similar to the other two kits, JBS urine and plasma cfDNA extraction can be automated using the JBS JPurX-S200 instrument to reduce variation due to manual processing. While only normal donors were included in this study, extraction of urine cfDNA from donors with various diseases, such as cancer [16,17] and infection-related conditions [16], may be hampered by the even greater biospecimen variability associated with such pathologies. The effects of this variability on the performance of any new cfDNA isolation method will need to be thoroughly assessed.

In summary, this study brings forth the importance of cfDNA isolation methodology in achieving high ccffDNA quality and the potential value of the PEG-based DNApure procedure in mitigating the impact of biospecimen variability on analytical performance. By improving the quality and yield of isolated ccffDNA, downstream analyses can also be significantly improved and standardized, enhancing liquid biopsy utility and facilitating the adoption of cfDNA-based assays in the clinic.

## Figures and Tables

**Figure 1 diagnostics-11-00650-f001:**
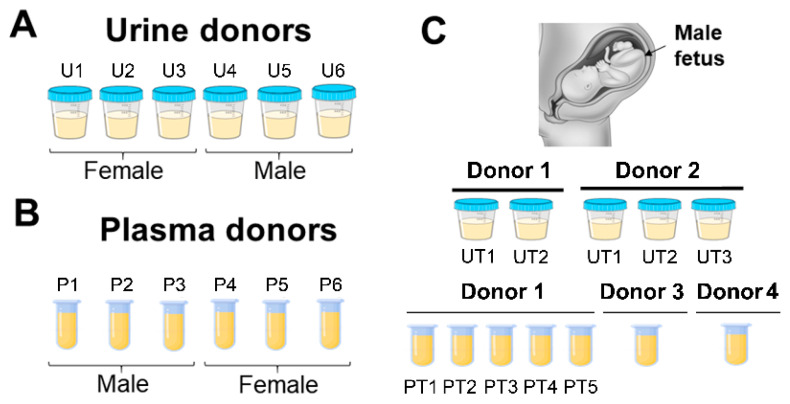
Normal donor specimens used in study. (**A**) Six normal urine donors (U1–6). (**B**) Six normal plasma donors (P1–6). Urine and plasma donors are non-matched. (**C**) Urine and plasma specimens collected from four third-trimester pregnant (PG) donors carrying a male fetus. Donors 1 (UT1, UT2) and 2 (UT1–3) urine was collected at five different timepoints. Donor 1 PG plasma was collected in-house at PT1–PT5 timepoints. Donors 3 and 4 PG plasma was collected by Lee BioSolutions. All urine samples were collected in-house.

**Figure 2 diagnostics-11-00650-f002:**
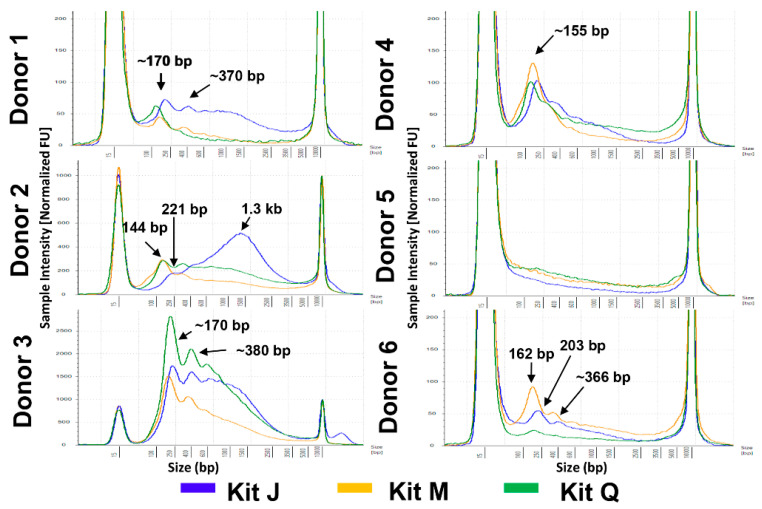
**** Fragment size profiles of cell-free DNA (cfDNA) extracted from urine of six normal donors. Representative profiles are shown for each donor after extraction following the manufacturers’ protocols. DNA input equivalents of 0.08 mL urine were loaded onto a D5000 High Sensitivity with ± 15% with a sizing accuracy of the screentape. All electropherograms are scaled to the sample. Kit J (blue), kit M (orange), and kit Q (green) cfDNA size profiles are indicated by color.

**Figure 3 diagnostics-11-00650-f003:**
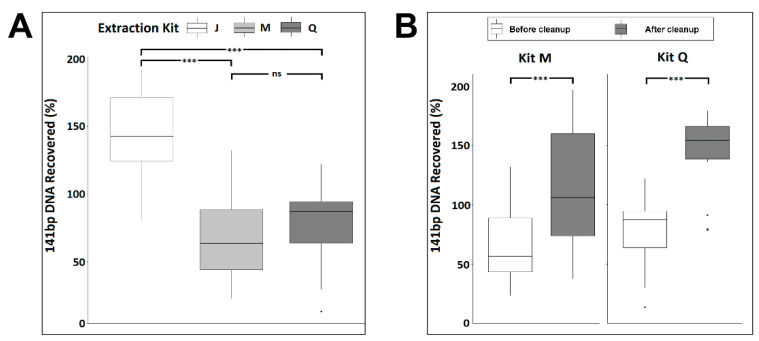
**** Synthetic 141 bp spike-in DNA recovery from normal urine donors by three cfDNA extraction kits as measured by qPCR before (**A**) and after (**B**) DNApure cleanup. For each kit, triplicate extractions were performed for each of the six urine donors. “Cleanup” indicates samples were further purified using DNApure cleanup. Significance levels of differences between samples under brackets as assessed with Tukey’s post hoc test are indicated (ns, *p* > 0.1; ***, *p* < 0.001).

**Figure 4 diagnostics-11-00650-f004:**
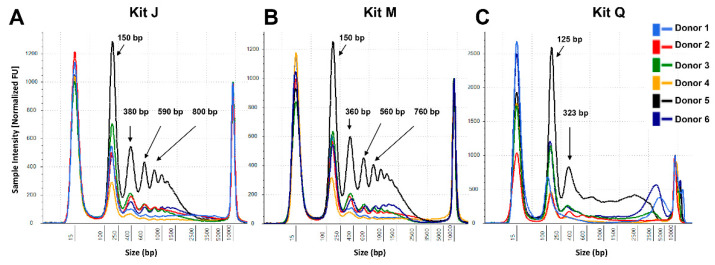
CfDNA size profiles of plasma samples extracted from six normal donors. Representative profiles are shown for each donor for kits J (**A**), M (**B**), and Q (**C**). The cfDNA extracted with kits M and Q was subjected to the DNApure cleanup procedure prior to TapeStation loading to remove gel impurities. DNA input equivalents of 0.08 mL plasma were loaded onto a D5000 High Sensitivity screentape. Averaged TapeStation software-identified DNA peaks are indicated.

**Figure 5 diagnostics-11-00650-f005:**
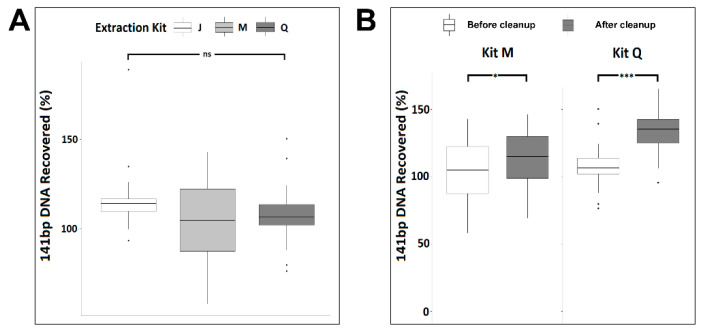
Synthetic 141 bp spike-in DNA recovery from six normal plasma donors by three cfDNA extraction kits as measured by qPCR before (**A**) and after (**B**) DNApure cleanup. For each kit, triplicate extractions were performed for each of the six plasma donors. “Cleanup” indicates samples were further purified using DNApure cleanup. Significance levels of differences between samples under brackets as assessed with Tukey’s post hoc test are indicated (ns, *p* > 0.1; *, <0.05; ***, *p* < 0.001).

**Figure 6 diagnostics-11-00650-f006:**
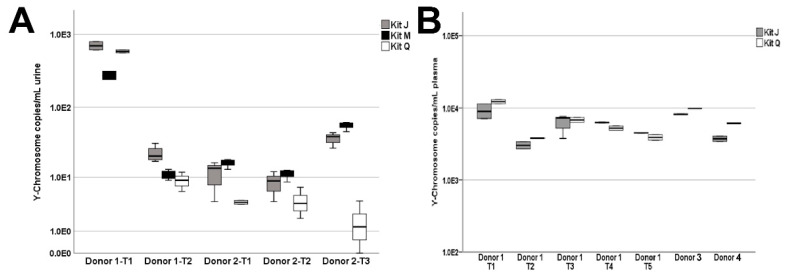
Y-chromosome DNA extracted from liquid biopsy specimens of pregnant donors carrying a male fetus. (**A**) Y-chromosome quantities isolated from urine collected from two donors over five time points (Donor 1: T1 and T2; Donor 2: T1, T2, and T3). (**B**) Y-chromosome quantities isolated from 0.5 mL plasma. Donor 1 had matched plasma sampled at times T1 and T2 and subsequently plasma only at times T3, T4, and T5. Due to limited PG plasma quantities, cfDNA was extracted with kits J and Q only.

## Data Availability

The data generated and analyzed during this study are available from the corresponding author on reasonable request.

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
