# Peer review of "A New Method for Improving Extraction Efficiency and Purity of Urine and Plasma Cell-Free DNA"

_diagnostics, 2021, doi:10.3390/diagnostics11040650_

Round 1

Reviewer 1 Report

Lin et al used three commercially available cfDNA extraction kits and then performed a PEG-based DNA cleanup procedure they call "DNApure" to see whether they could improve overall cfDNA quality and yield. They considered 6 normal donor urine and matched plasma samples, as well as the same fluids from 4 pregnant women carrying male fetuses. The 3 kits were: 1) the JBS cfDNA extraction kit, 2) the MagMAX Cell Free DNA Extraction kit and 3) the QIAamp Circulating Nucleic Acid Kit. Recovery was assessed by PCR product spike-in, endogenous TP53 plus Y-chromosome DNA analysis. They concluded that the JBS kit was the best in terms of recovery. Next, they report that the DNApure method significantly improved the amplification efficiency of spike-in DNA from urine and plasma. 

This is a well conducted study, and the results are clearly reported. I think the author should cite, briefly describe or speculate about another existing protocol for urine extraction (from Trovagene) which was start with up to 100ml input, prior to publication in this journal.

Author Response

Thank you for the suggestion. We have revised the discussion (lines 271-275, underlined) to further elaborate on the differences between the existing commercial urine extraction methods and the JBS method, which includes pre-processing centrifugation, and the impact of using silica columns compared to magnetic beads. Of note, Trovagene does not currently offer an isolation kit commercially, and the company has been re-branded. According to their previous publications (e.g., BRAF V600E mutations in urine and plasma cell-free DNA from patients with Erdheim-Chester disease), Trovagene’s urine cfDNA extraction procedure uses silica resin.

Reviewer 2 Report

1) I can not find the description for the reasonable sample size for this study, so the description would be needed.

2) The authors described that the paired t test or the Wilcoxon signed-rank test when parametric assumptions were not met. In results, the description for this aspect was not found. So the description for the statistical method for each results would be needed.         

Reviewer 3 Report

The manuscript by Lin et al described a new method to extract and purify celle-free DNA from urine and plasma. The manuscript is quite interesting. However, there are three major drawbacks:

1) the authors analyze only six samples and this cannot support definitive conclusion

2) the samples are all normal donors and in clinical practice liquid biopsy is used for some disease such as cancer

3) the three kit used for the comparison are for blood , but the authors stated that they analyzed urines.

The study must be improved, including an higher number of samples, samples from patients with several kind of disease (i.e. cancers, infectious disease) and using kit specific for urine or blood.

Reviewer 4 Report

The paper is well written. The methods are simple and effective.

I would accept in the present form.

Author Response

We appreciate reviewer’s recognition of this study.  Thank you!!

Round 2

Reviewer 3 Report

The authors addressed all the pointes and the manuscript can be accepted in the revised form.